# Primary cutaneous melanoma of the scalp: Patterns of clinical, histological and epidemiological characteristics in Brazil

**Ana Carolina Porto**[1]*, **Tatiana Pinto Blumetti**[1], **Ivan Dunshee de Abranches Oliveira Santos Filho**[1], **Vinicius Fernando Calsavara**[2], **João Pedreira Duprat Neto**[1], **Juliana Casagrande Tavoloni Braga**[1]

1 Cutaneous Oncology Department, A.C.Camargo Cancer Center, Sao Paulo, Brazil, 2 Center for Epidemiology and Statistics in Cancer, A.C.Camargo Cancer Center, Sao Paulo, Brazil

* carolsporto1@hotmail.com

**Data Availability Statement:** All relevant data are within the manuscript and its Supporting Information files.

## Abstract

### Background/Objectives

Scalp melanoma is a subgroup of melanomas on the head and neck, historically associated with worst prognosis. Knowledge of the usual presentation of scalp melanoma can help to understand the reasons for the poor outcomes of treatment. This is the first publication to describe the clinical, histopathological and epidemiological profile of patients with scalp melanoma in a Latin American population.

### Methods

A cross-sectional study was performed of all primary cutaneous melanoma seen by the A.C. Camargo Cancer Center between 2008 and 2018, using an electronic health records to access clinical and pathology data.

### Results

When compared to trunk and limbs, increasing age is expected for patients with scalp melanoma (10.865; CI (95%) = [8.303; 13.427]). Regarding risk of invasion, scalp melanomas have a higher chance to be invasive than in situ (OR = 1.783; CI (95%) = [1.196; 2.657]) and present with higher Breslow thickness (OR = 3.005; CI (95%) = [2.507; 3.601]). Scalp site was significantly associated with male sex (OR = 3.750; CI (95%) = [2.533; 5.554]), perineural invasion (OR = 13.739; CI (95%) = [5.919; 31.895]), ulceration (OR = 2.311; CI (95%) = [1.488; 3.588]), and mitosis (OR = 2.366; CI (95%) = [1.701; 3.292]), when compared to trunk and limbs melanoma.

### Conclusion

In the present study, head and neck melanomas represented 14.9% of all melanomas, a frequency slightly lower than that described in the literature and the mean age of melanoma on the scalp found was lower than that reported in the literature. These results could be explained by the demographic characteristics of Brazil, which has a population with a lower

**Funding:** The author(s) received no specific funding for this work.

**Competing interests:** The authors have declared that no competing interests exist.

life expectancy compared to the European and North American population. Scalp melanomas occurred in older men, were diagnosed with greater Breslow thickness and were associated with the presence of perineural invasion, mitosis and ulceration.

## Introduction

Twenty percent of melanomas are located on the head and neck [1], a higher frequency than expected, considering that this area represents only 9% of the body surface [2]. Melanomas on this location occur in older patients [3] and a higher incidence is expected with the increase in life expectancy of the population. Historically, they are associated with a greater risk of disease progression and death from melanoma than melanomas located elsewhere [4–7].

Melanomas located on the scalp are a subgroup that represent 35% of cases of head and neck melanomas, and are responsible for 5% of all melanomas. Scalp melanomas are characterized by a risk of death approximately 2 times greater than that of tumors located in the extremities, even after adjusting for Breslow thickness, age, sex and presence of ulceration [3,5,8]. Ozao-Choy *et al* showed that scalp location was an independent risk factor for worse overall survival and specific melanoma survival and concluded that scalp melanoma is a distinct entity and it is responsible for the worst prognosis associated with the head and neck melanoma group [2].

The reasons are uncertain [9], but some hypotheses cited to justify the worse prognosis of melanoma on this location are: hair coverage would result in a later diagnosis, high proportion of melanomas with rapid vertical growth, such as nodular and desmoplastic melanomas [10], higher blood and lymph flow [11,12] and the difficulty of obtaining adequate therapeutic surgical margins [2].

The gene profile signature in primary cutaneous melanoma appears to distinguish patients with low risk of metastatic recurrence from patients with high risk of recurrence. The presence of *BRAF* V600K mutation is associated with higher age, higher degrees of chronic sun damage and lower disease-free interval [13]. The possibly higher prevalence of V600K in scalp melanoma can be another explanation for its worst prognosis, but there is a lack of studies proving this association [10].

Knowledge of the usual presentation of scalp melanoma can help to understand the reasons for the worse prognosis on this location.

This study aimed to describe the clinical and histopathological profile of patients with melanoma on the scalp and to compare it with other areas of the head and neck, trunk and limbs.

## Methods

This is an observational, cross-sectional and retrospective study carried out at A.C.Camargo Cancer Center, São Paulo, Brazil, a tertiary clinical reference center. The study was determined to be exempt from needing informed consent and it was approved by the A.C.Camargo Cancer Center institutional Research Ethics Committee (2379/17). The study was conducted according to the Helsinki ethical principles

In the retrospective review of the electronic medical records from A.C.Camargo Cancer Center, São Paulo, Brazil, from January 2008 to December 2018, we identified all patients who had a histopathological diagnosis of melanoma.

Patients originated from public and private outpatient clinics in the skin cancer nucleus of A.C.Camargo Cancer Center, including non-referred and referred patients with a suspected or biopsy proven skin cancer.

The anatomical sites of primary cutaneous melanomas were classified into the following subgroups: scalp melanoma (SM), other areas in the head and neck region melanoma except scalp (HNM), trunk / limbs melanoma (TLM). For all patients, the following data were collected: sex, age at diagnosis, histopathological subtype of melanoma, presence of ulceration, mitosis, perineural invasion and association with nevus.

Descriptive analysis was performed to assess the clinical and histopathological characteristics of melanomas diagnosed in the studied period. Summary measures of position and dispersion, such as mean and standard deviation (SD), median and minimum and maximum, were considered for quantitative variables and absolute and relative frequencies (%) were used for qualitative variables.

To assess a possible association between two qualitative variables, the chi-square test was applied to the data. In order to compare the distribution of data for quantitative variables in relation to the three groups, the Kruskal-Wallis non-parametric test was used followed by multiple comparison (with Bonferroni correction) when the null hypothesis is rejected.

In order to identify if topography is an independent factor that explains the clinical and histopathological characteristics of melanoma, simple linear and logistic regression models were fitted to the data depending on the outcome variable. To assess if topography affects the Breslow thickness, a generalized linear model with Gamma distribution and log link function was fitted to the dataset. For the logistic regression the measure of association of interest is given by the odds ratio (OR) with a 95% confidence interval (CI (95%)). For all hypothesis tests, the level of significance was set at 5%. Thus, results from which p-values were less than 0.05 were considered statistically significant. IBM SPSS version 24 software was used in all data analyzes.

## Results

From January 2008 to January 2018, 3026 patients were diagnosed with melanoma at the institution. Of these, 453 (14.9%) were located on the head and neck region, including 154 (5.1%) located on the scalp. The clinical and histopathological characteristics of the melanomas diagnosed in the studied period and the results are described in Table 1.

Scalp melanomas occurred in significantly older patients (mean = 62 years, SD = 16 years) compared to trunk and limb melanoma (mean = 52 years, SD 15 = years), but at a similar age to patients with melanoma on other head and neck areas (mean = 59 years, SD 18 years).

Through the association test, we observed that topography is associated with sex (p <0.0001). Males represent 78,6% of patients with melanoma on the scalp, 49,6% of patients with melanoma on the trunk and limbs, 61,5% of patients with melanoma in the other head and neck areas.

A greater proportion of invasive melanomas was observed on the scalp (78.8%) in comparison to trunk and limbs (67.6%) and other head and neck areas (51.8%). When in situ melanomas were excluded, invasive melanomas on the scalp were significantly thicker than melanomas in the other groups (SM mean = 4.36mm, SD = 5.89mm; TLM mean = 1.45mm, SD = 2.29mm; HNM mean = 1.24mm, SD = 1.35mm). The desmoplastic and nodular histopathological subtypes, the presence of ulceration and perineural invasion were significantly associated with scalp location (p <0.001).

According to estimates from the fitted linear regression model (Table 2), patients with SM are on average 10.8 years older than patients with TLM (10.865; CI (95%) = [8.303; 13.427]). Patients with HNM are on average 7.6 years older than TLM patients (7.692; CI (95%) = [5.805; 9.579]).

From the simple logistic regression, scalp site was significantly associated with male sex (OR = 3.750; CI (95%) = [2.533; 5.554]), as well as others areas in the head and neck region (OR = 1.637; CI (95%) = [1.280; 2.092]), compared to trunk / limbs.

**Table 1. Clinical and histopathological characteristics of melanoma in relation to the scalp, head and neck and trunk/limbs.**

| Variable | | SM (n = 154) | HNM (n = 299) | TLM (n = 2567) | Value p* | Multiple comparison | |
|---|---|---|---|---|---|---|---|
| | | | | | | Group | Value p* adjusted |
| **Sex—n (%)** | Female | 33 (21.5) | 115 (38.5) | 1298 (50.6) | <0.0001* | | |
| | Male | 121 (78.6) | 184 (61.5) | 1269 (49.4) | | | |
| **Age (years)—mean (SD)** | | 62.48 (15.87) | 59.31(18.08) | 51.62 (15.46) | < 0.0001* | TLM vs SM | < 0.0001** |
| | | | | | | TLM vs HNM | < 0.0001** |
| | | | | | | SM vs HNM | 0.141** |
| **Breslow thickness—n (%)** | In situ | 32 (21.2) | 144 (48.2) | 830 (3.,4) | <0.0001* | | |
| | Invasive | 119 (78.8) | 155 (51.8) | 1731 (67.6) | | | |
| **Invasive Breslow thickness (mm)** | Mean (SD) | 4.36 (5.89) | 1.24 (1.35) | 1.45 (2.29) | <0.0001* | TLM vs SM | < 0.0001** |
| | Median (min—max) | 1.98 (0.22–40) | 0.75 (0.1–8) | 0.7 (0.1–26) | | TLM vs HNM | 0,999(m** |
| | | | | | | SM vs HNM | < 0.0001** |
| **Histopathological n (%)** | Superficial spreading | 78 (52.7) | 167 (65) | 1979 (86.2) | < 0.001* | | |
| | Lentiginous | 16 (10.8) | 65 (25.3) | 102 (4.4) | | | |
| | Nodular | 22 (14.9) | 9 (3.5) | 131 (5.7) | | | |
| | Desmoplastic | 12 (8.1) | 1 (0.4) | 2 (0.1) | | | |
| | Spitzoid | 2 (1.4) | 7 (2.7) | 33 (1.4) | | | |
| | Others | 18 (12.2) | 8 (3.1) | 49 (2.1) | | | |
| **Ulceration—n (%)** | Present | 27 (18.5) | 16 (5.5) | 222 (8.9) | < 0.001* | | |
| **Neurotropism -n (%)** | Present | 10 (6.8) | 3 (1.1) | 13 (0.5) | < 0.001* | | |
| **Mitosis n (%)** | Present | 77(51)) | 69 (23.8) | 762 (30.5) | < 0.001* | TLM vs SM | < 0.0001** |
| | | | | | | TLM vs HNM | 0.034** |
| | | | | | | SM vs HNM | < 0.0001** |
| **Nevus—n (%)** | Present | 18 (12.5) | 51 (18.4) | 540 (22.4) | < 0.001* | | |

*Teste de Kruskal-Wallis.

** Multiple comparison with Bonferroni correction.

SM, scalp melanoma; HNM, head and neck melanoma; TLM, trunk/limb melanoma; standard deviation (SD); min, minimum; max, maximum

SM have a higher chance to present ulceration in relation to TLM (OR = 2.311; CI (95%) = [1.488; 3.588]),]), while there were no difference in the occurrence of ulcerations on HNM in relation to TLM (OR = 0.595; CI (95%) = [0.353; 1.003]). SM have lower chance of being associated with nevi (OR = 0.495; CI (95%) = [0.299; 0.818]), while there were no difference in the association with nevi between HNM and TLM (OR = 0.782; CI (95%) = [0.568; 1.075]),

The SM have higher chance to present with mitosis (OR = 2.366; CI (95%) = [1.701; 3.292]) and perineural invasion (OR = 13.739; CI (95%) = [5.919; 31.895]), while HNM have a lower chance to present with mitosis (OR = 0.710; CI (95%) = [0.535; 0.943]) and no difference in the presence of perineural invasion (OR = 2.002; CI (95%) = [0.567; 7.070]), when compared to TLM.

Regarding Breslow thickness, SM have higher chance to be invasive (OR = 1.783; CI (95%) = [1.196; 2.657]) and, when invasive, to present with increased Breslow thickness (OR = 3.005; CI (95%) = [2.507; 3.601]). However, HNM have a higher chance to be in situ (OR = 0.516; CI (95%) = [0.405; 0.657]) and, when invasive, to present with lower Breslow thickness (OR = 0.858; CI (95%) = [0.732; 1.008]).

## Discussion

We describe one of the largest series of cases of scalp melanoma and its clinical and pathological characteristics, compared with melanomas located in other regions of the head and neck

**Table 2. Estimates of the parameters from simple linear, logistic and generalized regression models.**

| Linear regression model | | | | | |
| --- | --- | --- | --- | --- | --- |
| Outcome | Topography | Coefficient | 95% CI | | Value p |
| | | | Lower | Upper | |
| Age | TLM | Ref | | | |
| | SM | 10.865 | 8.303 | 13.427 | <0.0001 |
| | HNM | 7.692 | 5.805 | 9.579 | <0.0001 |
| Intercept | | 51.616 | 51.006 | 52.225 | <0.0001 |
| Simple Logistic Regression Model | | | | | |
| Outcome | Topography | OR | 95% CI | | p |
| | | | Lower | Upper | |
| Gender–Male | TLM | Ref | | | <0.0001 |
| | SM | 3.750 | 2.533 | 5.554 | <0.0001 |
| | HNM | 1.637 | 1.280 | 2.092 | <0.0001 |
| Ulceration–Yes | TLM | Ref | | | <0.0001 |
| | SM | 2.311 | 1.488 | 3.588 | <0.0001 |
| | HNM | 0.595 | 0.353 | 1.003 | 0.051 |
| Nevus–Yes | TLM | Ref | | | 0.009 |
| | SM | 0.495 | 0.299 | 0.818 | 0.006 |
| | HNM | 0.781 | 0.568 | 1.075 | 0.129 |
| Perineural invasion–Yes | TLM | Ref | | | <0.0001 |
| | SM | 13.739 | 5.919 | 31.895 | <0.0001 |
| | HNM | 2.002 | 0.567 | 7.070 | 0.281 |
| Invasive Melanoma (Breslow > 0mm)–Yes | TLM | Ref | | | <0.0001 |
| | SM | 1.783 | 1.196 | 2.657 | 0.004 |
| | HNM | 0.516 | 0.405 | 0.657 | <0.0001 |
| Mitosis–Yes | TLM | Ref | | | <0.0001 |
| | SM | 2.33 | 1.701 | 3.292 | <0.0001 |
| | HNM | 0.710 | 0.535 | 0.943 | 0.018 |
| Generalized linear model—Gamma distribution | | | | | |
| Outcome | Topography | OR | 95% CI | | p |
| | | | Lower | Upper | |
| Breslow thickness | TLM | Ref | | | |
| | SM | 3.005 | 2.507 | 3.601 | <0.0001 |
| | HNM | 0.859 | 0.732 | 1.008 | 0.062 |
| Intercept | | 1,453 | 1,387 | 1,521 | <0.0001 |

SM, scalp melanoma; HNM, head and neck melanoma; TLM, trunk/limb melanoma; standard deviation (SD); OR, odds ratio.

and melanomas on the trunk / limbs. In the present study, head and neck melanomas represented 14.9% of all melanomas, a frequency slightly lower than that described in the literature: 26.7% in a French regional study [2], 18% of invasive melanomas in the USA according to Surveillance, Epidemiology and End Results program [3], 15–24% of all melanomas in other European populations [1,14,15]. These results could be explained by the demographic characteristics of Brazil, which has a population with a lower life expectancy compared to the European and North American population [16].

The mean age of melanoma on the scalp found in our study was lower than that described in the literature (62 vs. 65 years, respectively) [17], a fact also justified by the lower life expectancy in the Brazilian population [16]. Patients with scalp melanoma were almost 11 years

older than patients with melanomas located on the trunk and limbs (62 vs. 51 years), but only 3 years older than patients with melanomas located on the other head and neck areas (62 vs. 59 years). These results are in agreement with other population studies, in which head and neck melanomas are more frequent in the elderly than in young people. Scalp melanomas occurred predominantly in males, six times more commonly in men than in women, which is consistent with other reports in the literature [10,18].

Scalp melanomas were more frequently invasive and associated with greater mean Breslow thickness than melanomas elsewhere in the head and neck and in the trunk and limbs (78.6% and 4.36mm; 51.8% and 1.24mm; 67.6% and 1.45mm, respectively), a finding reported in other studies [5]. Late diagnosis due to the fact that the tumor is hidden under the hair has been a justification often described for this greater Breslow thickness. Interestingly, androgenetic alopecia is highly prevalent in patients with scalp melanoma and Benati *et al* found thicker melanomas in bald patients compared to patients with hair, a fact that weakens the above justification [17].

Rapidly growing tumors appear to have a greater influence on tumor thickness than delayed diagnosis [19]. Nodular and desmoplastic melanomas are subtypes associated with rapid growth and some previous reports have shown that they occur more frequently on the scalp. Both are often amelanotic, more common in older age groups and diagnosed with greater Breslow thickness [5,10]. Nodular melanoma grows 0.49 mm/month of thickness and it accounts for nearly 50% of all melanomas thicker than 2 mm [20]. In our study, 14.9% of melanomas on the scalp were nodular, a proportion significantly higher than that observed in other areas. Desmoplastic melanoma is an uncommon type of melanoma, representing less than 5% of all melanomas, and it has a different clinical behavior, which includes a higher recurrence rate, and less propensity to metastasize to the regional lymph nodes [21]. Additionally, desmoplastic melanoma has a high frequency of *NF1* mutations, distinct from conventional melanoma [22]. In our study, 8.1% were desmoplastic melanoma, a proportion significantly higher than that observed in other head and neck areas and trunk and limb sites.

Compared with head/neck and trunk/limb melanomas, scalp melanomas were associated with greater Breslow thickness, mitosis, ulceration and perineural invasion, histological features associated with more aggressive tumors and a worse prognosis.

## Limitations

This study has some limitations, it is a retrospective observational analysis, so important data such as the presence of alopecia, some clinical characteristics and the pattern of sun exposure have not been evaluated. Knowledge of the pattern of sun exposure and the molecular profile of melanoma on the scalp will likely contribute to understanding the unique behavior of this subgroup of melanomas.

## Strengths

This is the first publication to describe scalp melanomas in a Latin American population. We believe that our study will contribute to the literature and that additional studies, which include the molecular profile of melanoma on the scalp, will be necessary to understand the unique behavior of this subgroup of melanomas.

## Conclusion

Melanoma on the scalp has distinct clinical and histopathological characteristics compared to melanoma from other head and neck areas and melanoma on the trunk / limbs. Scalp melanomas are more frequently invasive and diagnosed with greater Breslow thickness, presence of

ulceration, mitosis and perineural invasion. These characteristics may justify the more aggressive behavior of the tumor at that location. As it is associated with higher rates of mortality and recurrence, careful examination of the scalp should be part of the routine dermatological examination.

## Supporting information

**S1 Data.**
(XLSX)

## Acknowledgments

The authors would like to thank A.C. Camargo Cancer Center (São Paulo, Brazil), Department of Cutaneous Oncology.

## Author Contributions

**Conceptualization:** Ana Carolina Porto, Tatiana Pinto Blumetti, Ivan Dunshee de Abranches Oliveira Santos Filho.

**Data curation:** Ana Carolina Porto, Ivan Dunshee de Abranches Oliveira Santos Filho.

**Formal analysis:** Ana Carolina Porto, Ivan Dunshee de Abranches Oliveira Santos Filho.

**Funding acquisition:** Ana Carolina Porto.

**Investigation:** Ana Carolina Porto, Tatiana Pinto Blumetti.

**Methodology:** Ana Carolina Porto, Tatiana Pinto Blumetti, Vinicius Fernando Calsavara.

**Project administration:** Ana Carolina Porto, Vinicius Fernando Calsavara.

**Resources:** Ana Carolina Porto, Vinicius Fernando Calsavara.

**Software:** Ana Carolina Porto, Vinicius Fernando Calsavara.

**Supervision:** Ana Carolina Porto, Juliana Casagrande Tavoloni Braga.

**Validation:** Ana Carolina Porto, João Pedreira Duprat Neto, Juliana Casagrande Tavoloni Braga.

**Visualization:** Ana Carolina Porto, João Pedreira Duprat Neto, Juliana Casagrande Tavoloni Braga.

**Writing – original draft:** Ana Carolina Porto.

**Writing – review & editing:** Ivan Dunshee de Abranches Oliveira Santos Filho, João Pedreira Duprat Neto, Juliana Casagrande Tavoloni Braga.

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
