## [Decision Letter · Decision Letter 0]

3 Aug 2020

PONE-D-20-18647

Primary cutaneous melanoma of the scalp: patterns of clinical, histological and epidemiological characteristics in Brazil

PLOS ONE

Dear Dr. Porto,

Thank you for submitting your manuscript to PLOS ONE. After careful consideration, we feel that it has merit but does not fully meet PLOS ONE’s publication criteria as it currently stands. Therefore, we invite you to submit a revised version of the manuscript that addresses the points raised during the review process.

As you will see the reviewers have the opinion that your work is interesting and has merit. However a number of substantial improvements are required before approval of your work. Please address ALL the reviewers concerns in your revised Ms 

We look forward to receiving your revised manuscript.

Kind regards,

Paula Soares

Academic Editor

PLOS ONE

Journal Requirements:

2. Thank you for including your ethics statement: 'The project was approved by the institutional Research Ethics Committee (2379/17).'

(a) Please amend your current ethics statement to include the full name of the ethics committee/institutional review board(s) that approved your specific study. 

(b) Once you have amended this/these statement(s) in the Methods section of the manuscript, please add the same text to the “Ethics Statement” field of the submission form (via “Edit Submission”).

3. In the ethics statement in the manuscript and in the online submission form, please provide additional information about the patient records used in your retrospective study, including: a) whether all data were fully anonymized before you accessed them; b) the date range (month and year) during which patients' medical records were accessed; and c) the source of the medical records analyzed in this work (e.g. hospital, institution or medical center name). If patients provided informed written consent to have data from their medical records used in research, please include this information.

Reviewers' comments:

Reviewer's Responses to Questions

**Comments to the Author**

1. Is the manuscript technically sound, and do the data support the conclusions?

Reviewer #1: Yes

Reviewer #2: Yes

2. Has the statistical analysis been performed appropriately and rigorously? 

Reviewer #1: I Don't Know

Reviewer #2: I Don't Know

3. Have the authors made all data underlying the findings in their manuscript fully available?

Reviewer #1: Yes

Reviewer #2: Yes

4. Is the manuscript presented in an intelligible fashion and written in standard English?

Reviewer #1: Yes

Reviewer #2: Yes

5. Review Comments to the Author

Reviewer #1: The authors have submitted a single institution retrospective case series of scalp melanoma. The study is interesting and clinically relevant. Although the findings are not particularly novel or unexpected, they strengthen existing knowledge and are novel in terms of apparently being the first reported series from Brazil and Latin America overall. Important variables not reported include molecular features (e.g., BRAF V600E mutation status) and outcome/survival.

Although melanoma involving the scalp is well established, the high proportion of desmoplastic melanoma (DM) (compared to non-scalp, both NNM and TLM) might seem to mitigate adverse prognostic factors (DM perhaps with same or higher rate of local recurrence but lower rate of distant metastasis and death compared to nodular melanoma). DM is also genetically distinct from other forms of melanoma and can be subdivided into pure and mixed types. Can the authors comment on these aspects?

Please comment, if possible, regarding the significance, if any, of specific regions within the scalp. E.g., were photographs available? Was there a correlation with covered versus bald/alopecic scalp sites (e.g., Ref #19) The authors appear to acknowledge in Limitations that they did not evaluate this.

Abstract, 1st sentence: As worded, this sentence is not true (e.g. PMID 24281175 from 10 years prior). Please re-word to state that this is the 1st series from Latin America to focus on the unique clinical/histologic/epidemiological aspects of scalp melanoma.

Reviewer #2: These authors performed a retrospective analysis of all cutaneous melanomas diagnosed at a South American referral center between 2008-2018. They identify statistically-significant demographic and histopathologic differences comparing scalp melanomas to melanomas involving other anatomic locations. Specifically, they find that scalp melanoma patients are generally older and that the tumors are more likely to demonstrate adverse prognostic indicators. Their findings are generally consistent with findings reported in other studies, but this study adds additional information to the literature as it represents a geographic region and study population that has is historically underrepresented in the melanoma literature. I thank the authors for this work.

In your methods, please specify if your analysis included melanoma in situ (I believe it did). If so, I suggest you include data on MIS in Table 1 -- perhaps, show the distribution of all melanomas, MIS, and invasive disease. Page 9, lines 142-148 describe data on the variation in distribution of invasive melanomas -- I'd like see those numbers displayed in your table.

When you reference Breslow depth I recommend that you round to the nearest one tenth -- i.e., 3.43 should be 3.4. This aligns with current AJCC staging criteria.

I defer to the editors, but I believe Tables 2 and 3 could either be simplified or presented as supplemental material. Is talbe 2 necessary? Your point is, for example, that scalp melanomas have higher ORs for depth, perineural invasion, male sex, etc following regression modeling. Clinicians and pathologists will read this manuscript, and I suggest the analysis be explained in simplified terms (with more detail available).

Page 11, line 161 you state, "Increased Breslow thickness was identified as an important independent factor to explain the occurrence on the scalp.." I believe this statement is backwards. At the very least it is more biologically plausible that scalp location is a risk factor for higher Breslow depth. Further in your discussion you make similar statement regarding other variables (ulceration, mitoses, perineural invasion). It does not make sense that poor prognostic indicators "explain" scalp location but, rather, scalp location "explains" the presentation with poor prognostic indicators.

Do you have outcomes data based on anatomic location? I will not be surprised if the answer is 'no' -- outcomes data more more difficult to collect -- but if you have them, please report them (in this manuscript or another).

Your references require significant editing. Cursory review reveals multiple incomplete references (9, 12, 13).

6. PLOS authors have the option to publish the peer review history of their article (what does this mean?). If published, this will include your full peer review and any attached files.

Reviewer #1: No

Reviewer #2: No

---

## [Author Response · Author response to Decision Letter 0]

9 Sep 2020

PONE-D-20-18647

Primary cutaneous melanoma of the scalp: patterns of clinical, histological and epidemiological characteristics in Brazil

Dear Editor Paula Soares and reviewers,

We would like to thank you for the opportunity to submit our manuscript to the Plos One as well as for considering our paper as interesting for your readership. Referent to our manuscript (PONE-D-20-18647), the questions pointed by the reviewers are answered as follows: 

# Editor

Comment: 1. Please ensure that your manuscript meets PLOS ONE's style requirements, including those for file naming. The PLOS ONE style templates can be found at

Answer: Thank you for the instructions, we changed the text as necessary.

Comment: 2. Thank you for including your ethics statement: 'The project was approved by the institutional Research Ethics Committee (2379/17).'

(a) Please amend your current ethics statement to include the full name of the ethics committee/institutional review board(s) that approved your specific study. 

(b) Once you have amended this/these statement(s) in the Methods section of the manuscript, please add the same text to the “Ethics Statement” field of the submission form (via “Edit Submission”).

Answer: Thank you for the instructions, we included the full name of the ethics committee review board.

Comment: 3. In the ethics statement in the manuscript and in the online submission form, please provide additional information about the patient records used in your retrospective study, including: a) whether all data were fully anonymized before you accessed them; b) the date range (month and year) during which patients' medical records were accessed; and c) the source of the medical records analyzed in this work (e.g. hospital, institution or medical center name). If patients provided informed written consent to have data from their medical records used in research, please include this information.

Answer: Thank you for the instructions, we add this information in the manuscript

# Reviewers

Reviewer #1: 

Comment: “The authors have submitted a single institution retrospective case series of scalp melanoma. The study is interesting and clinically relevant. Although the findings are not particularly novel or unexpected, they strengthen existing knowledge and are novel in terms of apparently being the first reported series from Brazil and Latin America overall. Important variables not reported include molecular features (e.g., BRAF V600E mutation status) and outcome/survival.”

Answer: Thank you very much for your considerations. As it is a retrospective research, unfortunately these data were not available. 

Comment: “Although melanoma involving the scalp is well established, the high proportion of desmoplastic melanoma (DM) (compared to non-scalp, both NNM and TLM) might seem to mitigate adverse prognostic factors (DM perhaps with same or higher rate of local recurrence but lower rate of distant metastasis and death compared to nodular melanoma). DM is also genetically distinct from other forms of melanoma and can be subdivided into pure and mixed types. Can the authors comment on these aspects?”

Answer: Thank you for the instructions, we comment with more details, between lines 197 and 207, the specific behavior of nodular and desmoplastic melanomas.

Comment: “Please comment, if possible, regarding the significance, if any, of specific regions within the scalp. E.g., were photographs available? Was there a correlation with covered versus bald/alopecic scalp sites (e.g., Ref #19) The authors appear to acknowledge in Limitations that they did not evaluate this.”

Answer: As it is a retrospective study, the specific location within the scalp was not described in the vast majority of medical records evaluated. And photographs were also unavailable in almost all cases.

Comment: “Abstract, 1st sentence: As worded, this sentence is not true (e.g. PMID 24281175 from 10 years prior). Please re-word to state that this is the 1st series from Latin America to focus on the unique clinical/histologic/epidemiological aspects of scalp melanoma.”

Answer: The article PMID 24281175, published 2010, affirmed that the poor prognosis for scalp melanoma is due to late diagnosis, as the scalp region is often covered by hair. Meanwhile, in a more recent article published in 2018, PMID: 28543136, on multivariate analysis, scalp location was an independent predictor of worse melanoma-specific and overall survival. And the authors concluded that the pathophysiology of this poor prognosis remains to be determined and is not due to the late diagnosis. However, we re-word the 1st sentence as suggested.

Reviewer #2: 

Comment:” These authors performed a retrospective analysis of all cutaneous melanomas diagnosed at a South American referral center between 2008-2018. They identify statistically-significant demographic and histopathologic differences comparing scalp melanomas to melanomas involving other anatomic locations. Specifically, they find that scalp melanoma patients are generally older and that the tumors are more likely to demonstrate adverse prognostic indicators. Their findings are generally consistent with findings reported in other studies, but this study adds additional information to the literature as it represents a geographic region and study population that has is historically underrepresented in the melanoma literature. I thank the authors for this work.

Answer: All the authors would really like to thank your comment.

Comment: “In your methods, please specify if your analysis included melanoma in situ (I believe it did). If so, I suggest you include data on MIS in Table 1 -- perhaps, show the distribution of all melanomas, MIS, and invasive disease. Page 9, lines 142-148 describe data on the variation in distribution of invasive melanomas -- I'd like see those numbers displayed in your table.”

Answer:: Yes, we included melanoma in situ in our analysis. However, for the calculation of the average thickness of Breslow, only invasive melanomas were included. We make this clearer in the table and in the text, as you suggested

Comment: “When you reference Breslow depth I recommend that you round to the nearest one tenth -- i.e., 3.43 should be 3.4. This aligns with current AJCC staging criteria.

Answer: We update the Breslow thickness as you suggested.

Comment: “I defer to the editors, but I believe Tables 2 and 3 could either be simplified or presented as supplemental material. Is talbe 2 necessary? Your point is, for example, that scalp melanomas have higher ORs for depth, perineural invasion, male sex, etc following regression modeling. Clinicians and pathologists will read this manuscript, and I suggest the analysis be explained in simplified terms (with more detail available).

Answer: We appreciate and agree with this suggestion. As we had to invert the order of the variables requested in the comment below, we had to adapt the statistical model and we have simplified it to just one table.

Comment: “Page 11, line 161 you state, "Increased Breslow thickness was identified as an important independent factor to explain the occurrence on the scalp.." I believe this statement is backwards. At the very least it is more biologically plausible that scalp location is a risk factor for higher Breslow depth. Further in your discussion you make similar statement regarding other variables (ulceration, mitoses, perineural invasion). It does not make sense that poor prognostic indicators "explain" scalp location but, rather, scalp location "explains" the presentation with poor prognostic indicators.

Answer: We also appreciate and agree with this suggestion However, in order to reverse all the statements, the statistical analysis needed to be modified. Through this new analysis, the results and conclusions were maintained without significant changes.

Comment: “Do you have outcomes data based on anatomic location? I will not be surprised if the answer is 'no' -- outcomes data more more difficult to collect -- but if you have them, please report them (in this manuscript or another).”

Answer: Unfortunately we don’t have these data.

Comment: “Your references require significant editing. Cursory review reveals multiple incomplete references (9, 12, 13).

Answer: We update the references.

---

## [Decision Letter · Decision Letter 1]

29 Sep 2020

PONE-D-20-18647R1

Primary cutaneous melanoma of the scalp: patterns of clinical, histological and epidemiological characteristics in Brazil

PLOS ONE

Dear Dr. Porto,

Thank you for submitting your manuscript to PLOS ONE. After careful consideration, we feel that it has merit but does not fully meet PLOS ONE’s publication criteria as it currently stands. Therefore, we invite you to submit a revised version of the manuscript that addresses the points raised during the review process.

Thank you for the revisions done in the Ms. However some minor points remain to be revised.

Please addresses the few points raised by Reviewer 1.

Thank you

We look forward to receiving your revised manuscript.

Kind regards,

Paula Soares

Academic Editor

PLOS ONE

Reviewers' comments:

Reviewer's Responses to Questions

**Comments to the Author**

1. If the authors have adequately addressed your comments raised in a previous round of review and you feel that this manuscript is now acceptable for publication, you may indicate that here to bypass the “Comments to the Author” section, enter your conflict of interest statement in the “Confidential to Editor” section, and submit your "Accept" recommendation.

Reviewer #1: (No Response)

Reviewer #2: All comments have been addressed

2. Is the manuscript technically sound, and do the data support the conclusions?

Reviewer #1: Yes

Reviewer #2: Yes

3. Has the statistical analysis been performed appropriately and rigorously? 

Reviewer #1: I Don't Know

Reviewer #2: Yes

4. Have the authors made all data underlying the findings in their manuscript fully available?

Reviewer #1: Yes

Reviewer #2: Yes

5. Is the manuscript presented in an intelligible fashion and written in standard English?

Reviewer #1: Yes

Reviewer #2: Yes

6. Review Comments to the Author

Reviewer #1: Thank you for responding to nearly all of the Reviewer comments with this improved, revised manuscript.

Line 197: *It would still be useful to address, at least generally, whether the desmoplastic melanomas (DM) in this series were pure vs mixed type. PMID 23267722

Line 197: *The authors have not provided convincing data to support their assertion that DM is a rapidly growing subset of melanoma. Mitotic rate may be reasonably used as a surrogate marker for tumor growth, and although scalp melanoma overall may grow relatively rapidly, especially nodular melanoma, DM had a lower mitotic rate in PMID 25142970. Suggest removing or else providing support for this assertion.

Reviewer #2: (No Response)

7. PLOS authors have the option to publish the peer review history of their article (what does this mean?). If published, this will include your full peer review and any attached files.

Reviewer #1: No

Reviewer #2: No

---

## [Author Response · Author response to Decision Letter 1]

30 Sep 2020

PONE-D-20-18647

Primary cutaneous melanoma of the scalp: patterns of clinical, histological and epidemiological characteristics in Brazil

Dear Editor Paula Soares and reviewers,

We would like to thank you for the opportunity to submit our manuscript to the Plos One as well as for considering our paper as interesting for your readership. Referent to our manuscript (PONE-D-20-18647), the minor questions pointed by the Reviewer #1 are answered as follows: 

# Reviewer #1: 

Comment: “It would still be useful to address, at least generally, whether the desmoplastic melanomas (DM) in this series were pure vs mixed type. PMID 23267722

Answer: Thank you very much for your suggestions, these considerations has improved our manuscript. 

All the scalp desmoplastic melanomas were pure type. We specified this in line 139 and 140.

Comment: “Line 197: *The authors have not provided convincing data to support their assertion that DM is a rapidly growing subset of melanoma. Mitotic rate may be reasonably used as a surrogate marker for tumor growth, and although scalp melanoma overall may grow relatively rapidly, especially nodular melanoma, DM had a lower mitotic rate in PMID 25142970. Suggest removing or else providing support for this assertion.

Answer: Thank you for the instructions, we detailed in line 198 – 201 that nodular melanoma is associated with rapidly growing, while desmoplastic melanoma is associated with higher rate of local recurrence.

---

## [Editor Report · Decision Letter 2]

5 Oct 2020

Primary cutaneous melanoma of the scalp: patterns of clinical, histological and epidemiological characteristics in Brazil

PONE-D-20-18647R2

Dear Dr. Porto,

We’re pleased to inform you that your manuscript has been judged scientifically suitable for publication and will be formally accepted for publication once it meets all outstanding technical requirements.

Kind regards,

Paula Soares

Academic Editor

PLOS ONE

---

## [Editor Report · Acceptance letter]

15 Oct 2020

PONE-D-20-18647R2 

Primary cutaneous melanoma of the scalp: patterns of clinical, histological and epidemiological characteristics in Brazil 

Dear Dr. Porto:

I'm pleased to inform you that your manuscript has been deemed suitable for publication in PLOS ONE. Congratulations! Your manuscript is now with our production department. 

Kind regards, 

on behalf of

Dr. Paula Soares 

Academic Editor

PLOS ONE